# Differences of Uric Acid Transporters Carrying Extracellular Vesicles in the Urine from Uric Acid and Calcium Stone Formers and Non-Stone Formers

**DOI:** 10.3390/ijms231710010

**Published:** 2022-09-02

**Authors:** Zhijian Lin, Muthuvel Jayachandran, Zejfa Haskic, Sanjay Kumar, John C. Lieske

**Affiliations:** 1School of Chinese Materia Medica, Beijing University of Chinese Medicine, Beijing 100029, China; 2Division of Nephrology and Hypertension, Mayo Clinic College of Medicine, 200 1st Street SW, Rochester, MN 55905, USA; 3Division of Hematology Research, Mayo Clinic College of Medicine, 200 1st Street SW, Rochester, MN 55905, USA; 4Department of Physiology & Biomedical Engineering, Mayo Clinic College of Medicine, 200 1st Street SW, Rochester, MN 55905, USA; 5Department of Life Science, School of Basic Sciences and Research, Sharda University, Greater Noida 201310, Uttar Pradesh, India; 6Department of Laboratory Medicine and Pathology, Mayo Clinic College of Medicine and Science, 200 First Street SW, Rochester, MN 55905, USA

**Keywords:** urine pH, urinary vesicles, renal epithelial cells, nephrolithiasis, aciduria

## Abstract

Background: Low urine pH and volume are established risk factors for uric acid (UA) stone disease (UASD). Renal tubular epithelial cells exposed to an acidic pH and/or UA crystals can shed extracellular vesicles (EVs) into the tubular fluid, and these EVs may be a pathogenic biomarker of UASD. Methods: Urinary EVs bearing UA transporters (SLC2A9, SLC17A3, SLC22A12, SLC5A8, ABCG2, and ZNF365) were quantified in urine from UA stone formers (UASFs), calcium stone formers (CSFs), and age-/sex-matched non-stone formers (NSFs) using a standardized and published method of digital flow cytometry. Results: Urinary pH was lower (*p* < 0.05) and serum and urinary UA were greater (*p* < 0.05) in UASFs compared with NSFs. Urinary EVs carrying SLC17A3 and SLC5A8 were lower (*p* < 0.05) in UASFs compared with NSFs. Urinary EVs bearing SLC2A9, SLC22A12, SLC5A8, ABCG2, and ZNF365 were lower (*p* < 0.05) in CSFs than UASFs, while excretion of SLC17A3-bearing EVs did not differ between groups. Conclusion: EVs bearing specific UA transporters might contribute to the pathogenesis of UASD and represent non-invasive pathogenic biomarkers for calcium and UA stone risk.

## 1. Introduction

The incidence and prevalence of urinary stone disease (USD) appears to be increasing worldwide [1]. In the United States, uric acid (UA) stone disease (UASD) constitutes approximately 8–10% of total stone formers [2,3]. The most important metabolic risks for UASD include a low urine pH (<5.5) and low urine volume, or a combination of both [4,5]. Low urinary pH increases the risk of UASD because the acid form of UA (pKa 5.4) is poorly soluble [6]. In the kidney, filtered UA undergoes a complex series of reabsorption and secretion along the proximal nephron, the result ultimately determining urinary UA excretion [7,8]. A body of evidence, including human genome-wide association studies, suggests that the urate transporters SLC2A9 (solute carrier family 2, glucose transporter member 9), ABCG2 (ATP-binding cassette transporter G2), SLC17A3 (solute carrier family 17 member 3; a voltage-driven transporter excreting intracellular urate and organic anions from the blood into renal tubule cells), SLC22A12 (solute carrier family 22 member 12; organic anion/cation transporter), urate transporter 1 (URAT1), and SLC5A8 (solute carrier family member 8; an electrogenic sodium (Na^+^) and chloride (Cl^−^)-dependent sodium-coupled solute transporter) together regulate blood and urine UA concentrations [9,10,11,12,13]. ZNF365 (zinc finger protein 365) has also been associated with UA stones in both children and adults [14,15], and is thought to play a role in the expression of renal UA transporters. The relative expression of UA transporters and their proposed functions within the proximal and distal tubule are presented in Figure 1.

Extracellular vesicles (EVs) act as important mediators in normal physiology as well as many pathological states [16,17]. After release from parent cells, EVs affect the function of recipient cells via delivery of biologically active molecules that include proteins, lipids, and nucleic acids (DNA, mRNAs, and miRNAs) [18]. Recent studies have indicated that EVs can be used for the diagnosis, prognostic assessment, and management of individuals with suspected renal diseases [19]. Specific populations of urinary EVs appear to reflect underlying renal cell processes that might be associated with USD [20,21]. However, the pathophysiological association between EVs bearing UA transporters and UA stone formation has not been investigated.

The present study was designed to test the hypotheses that urinary EVs might carry renal UA transporters as signaling molecules, and urinary EV populations carrying specific UA transporters may differ in UA stone formers (UASFs) as a group. Thus, specific populations of urinary EVs bearing UA transporters could represent a biomarker of UA stone risk and provide clues regarding pathogenic steps. To test these hypotheses, we quantitated urinary EVs that contained six renal transporters associated with UA stone risk in cell-free urine samples from UASFs, calcium stone forms (CSFs), and age-/sex-matched non-stone formers (NSFs).

## 2. Methods

### 2.1. Study Participants

USD patients and non-stone forming controls were recruited at the Mayo Clinic O’Brien Urology Research Center from participants in IRB: 08-006541 (Epidemiology of nephrolithiasis and chronic kidney disease: Prospective Cohort) and IRB: 09-002083 (CT and Urinary Correlates of Renal Stone Precursor Lesions). Blood and 24 h urine samples (collected using toluene as a preservative) were obtained at the time of a Mayo Stone Clinic visit (UASFs and CSFs) or study visit (NSFs). Aliquots of the 24 h urine samples were centrifuged at 2100× *g*/3000 rpm for 10 min at 4 °C to remove the cells and larger molecular weight protein aggregates before freezing to −80 °C for future analysis [20,21].

All UASFs and CSFs had a complete evaluation in the Mayo Stone Clinic and those with secondary causes (enteric hyperoxaluria, primary hyperoxaluria, primary hyperparathyroidism, or renal tubular acidosis) were excluded. CSFs had stones composed of a majority calcium oxalate or hydroxyapatite. UASFs had a stone composed of a majority UA [20]. Age-/sex- matched NSFs were recruited form the general population. Stored urine samples from 18 UASFs (11 men and 7 women), 26 CSFs (16 men and 10 women), and 65 NSFs (39 men and 26 women) were used in the current study. This study was approved by the Institutional Review Board at Mayo Clinic, Rochester, MN, USA and all participants gave written informed consent for future analysis.

### 2.2. Laboratory Measurements

Clinical laboratory analyses were performed in the Clinical Laboratory Improvement Amendments (CLIA)-certified Mayo Clinic Renal Testing Laboratory, Rochester, MN, USA. Serum (uric acid, calcium, phosphorus, creatinine) and urine (sodium, potassium, chloride, uric acid, calcium, magnesium, phosphorus, citrate (via citrate lyase), oxalate (via oxalate oxidase)) biochemistries were analyzed using a Roche/Cobas C501 Autoanalyzer (Roche Diagnostics, Indianapolis, IN, USA); pH using a pH meter; and osmolality using a freezing point osmometer. The EQUIL2 computer program was utilized to calculate urinary supersaturations [22]. Serum and urine creatinine were measured using an isotope dilution mass spectrometry (IDMS)-traceable enzymatic creatinine assay (Roche Diagnostics, Indianapolis, IN, USA), and the serum creatinine-based Chronic Kidney Disease Epidemiology Collaboration (CKD-EPI) 2009 equation was used to determine estimated glomerular filtration rate (eGFR) [20,21].

### 2.3. Quantification of Extracellular Vesicles (EVs) by Digital Flow Cytometer

Frozen (−80 °C) cell-free urine samples were thawed in a 37 °C waterbath for 5 min prior to EV analysis by digital flow cytometry (BD FACSCanto™) using a size between 200 nm to 1000 nm in diameter and annexin-V positivity using the cross-validated protocol previously described by our group [20,21,23,24]. Previous analysis suggests EVs are stable under these conditions [20]. This analysis protocol detects both types of EVs, exosomes and microvesicles, carrying one of the six candidate UA transporter biomarkers. It is not possible to accurately differentiate between these two types of vesicles based on currently available surface markers, and evidence suggests both types of EVs have similar and overlapping bioactivities [25]. The absolute EV count either in the absence or presence of single or dual fluorophore-conjugated antibody staining was calculated using a previously validated method [20,21,23,24]. The number of EVs/µL urine was used to calculate EV excretion per 24 h and also normalized to EVs/mg urine creatinine [20,21,24,26,27,28]. All antibodies were fully validated with reagent controls and appropriate isotype control antibodies conjugated with the same fluorophores together with negative controls, as per our previous publications [20,21,23,24]. To prevent antibody–antibody interactions or binding, each fluorophore-conjugated antibody was used with and without dilutions with reagents only (without samples) and quantified by flow cytometer for optimization of antibodies, in order to prevent non-specific binding during quantification. We have also validated non-specific binding of each antibody using dose-response relationships in cell-free urine sample from the same patient or control.

Candidate EV-associated proteins for the current study were chosen from those that have been demonstrated to be, or might plausibly represent, a monogenic cause of UASD via literature review and the online database Genetics Home Reference (https://ghr.nlm.nih.gov/ accessed on 1 November 2018), and then narrowed down to those associated with urinary UA excretion or pH regulation: SLC2A9, SLC17A3, SLC22A12, SLC5A8, ABCG2, and ZNF365 (Figure 1) [9,10,11,12,13,14,15].

### 2.4. Chemicals, Reagents, and Antibodies

Fluorescein isothiocyanate (FITC) and phycoerythrin (PE)-conjugated recombinant annexin-V protein (catalog#: 55419-FITC; 563544-PE), mouse anti-human CD63 antibody conjugated with PE (catalog#: 556421), and TruCOUNT™ (4.2 μm) beads were obtained from BD Biosciences (San Jose, CA, USA); FITC-conjugated rabbit anti-human SLC2A9 (catalog#: abx316565), ZNF365 (catalog#: abx313505), and SLC17A3 (catalog#: abx305326) antibodies from Abbexa Ltd. (Cambridge, UK); FITC-conjugated rabbit anti-human SLC22A12 (catalog#: bs-10357R-FITC) from Bioss (Boston, MA, USA); Alex Fluor488-conjugated mouse anti-human SLC5A8 (catalog#: FAB8398G) antibody from R&D Systems (Minneapolis, MN, USA); and PE-conjugated mouse anti-human ABCG2 (catalog#: 332008) antibody from BioLegend (San Diego, CA, USA). HEPES and Hanks’ balanced salts were purchased from Sigma Chemicals Co., St. Louis, MO, USA. All other reagents and solvents used in this study were of analytical/reagent grade.

### 2.5. Data Analysis

Baseline clinical characteristics and serum and urine biochemical data were presented as the median with interquartile range for continuous variables, and number [percentage] for categorical variables of NSFs, UASFs, and CSFs. The numbers of EVs bearing UA transporters were presented on a log_10_ scale due to the range and non-normal distribution of the data. The statistical significance of differences between groups were examined using a non-parametric Wilcoxon rank sum test for continuous variables and Chi-square (Fisher’s exact) test for categorical variables using JMP software (SAS, Cary, NC, USA). *p* <0.05 was considered a statistically significant difference between groups.

## 3. Results

### Baseline Clinical Parameters

Age, sex, and systolic blood pressure; percentage of persons with malabsorption syndrome (but without enteric hyperoxaluria) or hypertension; serum calcium; and urine volume were all similar between groups (Table 1). Body mass index (BMI), serum UA, and creatinine; urinary excretions of protein, UA, phosphorous, and creatinine; and urinary supersaturation of UA were significantly greater in UASFs vs. NSFs (*p* < 0.05), whereas diastolic blood pressure, estimated glomerular filtration rate (eGFR), and urine pH were significantly lower (*p* < 0.05) in UASFs vs. NSFs (Table 1). Serum UA, phosphorous, and creatinine; urinary excretions of UA, phosphorous, calcium, and creatinine; and urinary supersaturation of calcium oxalate and UA were all significantly greater, while eGFR, and urine pH and osmolality were significantly lower in CSFs compared with NSFs (Table 1). Only the urinary excretion of calcium was significantly higher in CSFs compared with UASFs (Table 1).

Representative scatter and fluorescence dot plots of a single subject from the FACSCanto™ flow cytometer are shown in Figure 2. This particular subject had a significant number of urinary EVs carrying SLC2A9, SLC17A3, and ZNF365, but not SLC22A12 or SLC5A8. In the entire cohort, compared with NSFs, urinary excretion of EVs carrying SLC17A3 and SLC5A8 were significantly lower (*p* < 0.05), while urinary excretion of EVs carrying SLC2A9, SLC17A3, SLC22A12, ABCG2, and ZNF365 trended lower in UASFs (Figure 3). Urinary excretion of EVs bearing SLC2A9, SLC17A3, SLC22A12, SLC5A8, ABCG2, and ZNF365 were significantly lower in CSFs compared with NSFs (*p* < 0.05, Figure 3). Urinary excretion of EVs carrying SLC2A9-, SLC22A12-, SLC5A8-, ABCG2-, and ZNF365 were lower in CSFs compared with UASFs, while SLC17A3-bearing EVs did not differ between these two groups (Figure 3). Trends were similar when data for these six biomarker-expressing EVs were expressed as EVs excreted per 24 h, with numbers being lowest in CSFs, while UASFs were intermediate and closer to NSFs (Table 2).

## 4. Discussion

Low urine pH and urine volume are the major factors affecting urinary UA supersaturation (SS) and risk of UA stone formation [4,5,6,29,30,31]. In this study, urine pH and eGFR were lower, whereas BMI, serum UA, and creatinine; and urinary excretion of protein, UA, phosphorous, and creatinine, and urinary UA supersaturation were all greater in UASFs compared with NSFs. These findings are as expected from previous studies [31].

Tubular fluid pH and/or uric acid concentration, both risks for UA crystallization, could potentially be altered by EVs carrying transporters. In this study, urinary EVs carrying SLC17A3 and SLC5A8 were significantly lower in UASFs compared with NSFs. Urinary EVs carrying SLC2A9, SLC22A12, SLC5A8, ABCG2, and ZNF365 were lower in CSFs than UASFs, while EVs carrying SLC17A3 were not different between these two groups. SLC17A3 is a voltage-driven transporter of intracellular urate and organic anions from blood into renal tubular epithelial cells, and it thus plays an important role in metabolic disorders including serum uric acid concentration. Our current results suggest that renal tubular cells might respond to a specific type(s) of urinary crystals and tubular fluid compositions and release EVs containing specific UA transporters, and those changes may in turn reflect underlying UASD risk. These EVs could also potentially serve as a signaling mechanism that mediates UA stone risk (positively or negatively). For example, increasing tubular fluid uric acid concentration might downregulate proximal tubular cell secretion of EVs carrying SLC17A3 and SLC5A8 as a negative regulatory or unknown mechanism.

EVs can serve as potential mediators of intercellular signal communication via their bioactive molecules (proteins, lipids, and nucleic acids) [16,17,18,25]. Published studies suggest that the interaction between kidney epithelial cells and UA crystals could play a critical role in UA nephrolithiasis [32,33,34]. In vivo tubular fluid UA crystals could conceivably induce renal tubular epithelial cell injury, triggering EV secretion and initiating cellular processes related to UA stone formation. It is of note that in the current study, specific urinary EVs bearing UA-relevant transporters were reduced in CSFs compared with both NSFs and UASFs. The relative decrease of urinary EVs carrying UA transporters among CSFs compared with UASFs may reflect different degrees of ongoing UA crystal exposure that occur in vivo in these two groups of patients. The decrease in urinary excretion of EVs bearing UA transporters might reflect inhibition of EV excretion from tubular cells by urinary uric acid crystals and low urinary pH, since the urine pH of CSFs and UASFs were both reduced compared with controls, with the low urinary pH more pronounced among the UASF group. Furthermore, our study used the CSFs as a comparison group to analyze whether changes in urinary EV populations were a marker of stone former status in general, or more specific to the UASF group.

EV surface expression of SLC2A9, SLC5A8, SLC17A3, SLC22A12, ABCG2, and ZNF365 might reflect altered UA transport capacity of tubular epithelial cells in a given individual, and could also signal other distally located tubular epithelial cells. The populations of urinary EVs carrying SLC2A9, SLC17A3, and ZNF365 were greater than SLC22A12, SLC5A8, and ABCG2, and differed between UASFs and CSFs. These results suggest that urinary EVs could potentially represent pathogenic biomarkers of CSF vs. UASF risk. Thus, the use of EVs as noninvasive diagnostic biomarkers would enhance their clinical applicability.

This study has certain limitations. Only six pre-selected UA-transporters were measured in urine. This study also analyzed CSFs as a group, which is clinically relevant and commonly done but also represents a phenotypically diverse group. More evidence and mechanistic studies are needed to further define the precise relationship between urinary EVs carrying different transporters and stone pathogenesis. Furthermore, given the limited number of UASFs, we were not powered to do multivariable analysis and take into account the possible independent effect of urinary composition (as opposed to stone former type) on EV numbers. The candidate EV proteins were detected using flow cytometry with fluorophore-conjugated antibodies. In the future, it would be optimal to rigorously validate the expression of protein markers on purified urinary EVs using quantitative Western blot and/or mass spectrometry. Furthermore, the number of UASFs was relatively modest in comparison to the number of CSFs. The EV candidate biomarker pool could also potentially be increased, perhaps using an unbiased approach, such as by systematic bioinformatic analysis, and not only assessing EV-associated proteins but also bioactive nucleic acids (miRNA, mRNA, and DNA). Nevertheless, this is the first study to assess UA transporter-carrying EV populations in human urine from UASFs, CSFs, and NSF controls.

## 5. Conclusions

This study demonstrated that reduced urinary excretion of EVs carrying UA transporters by UASFs and CSFs might reflect the lower urine pH, volume, and resulting increased number of intratubular UA crystals that are associated with these two disease groups. The varied populations of EVs between disease groups might also reflect ongoing cell–cell communication. For example, renal UA transporter-carrying EVs could merge into a target cell membrane of recipient cells, which could in turn alter UA absorption and secretion, or otherwise alter the expression of this family of UA-transporters through mechanisms yet to be determined (e.g., EV-associated mRNA or miRNA, lithogenic proteins or UA nanocrystals). This process could protect against further renal injury when cells in one part of the nephron are exposed to low pH and/or UA crystals. If the physiological processes resulting in a lower urinary pH and/or hyperuricosuria and low urine flows are pronounced, the mechanisms that physiologically alter EV excretion from renal tubular cells might be overcome. Cellular responses to the UA and COM crystals could also play a role in the progression of chronic kidney disease that has been associated with hyperuricemia [35], and nephrolithiasis [36]. More mechanistic validation studies are needed to define the pathogenic role(s) of UA transporter-carrying urinary EVs in UA and calcium stone formation and their interrelationships, crystal nephropathy, and associated chronic kidney disease risk.

## Figures and Tables

**Figure 1 ijms-23-10010-f001:**
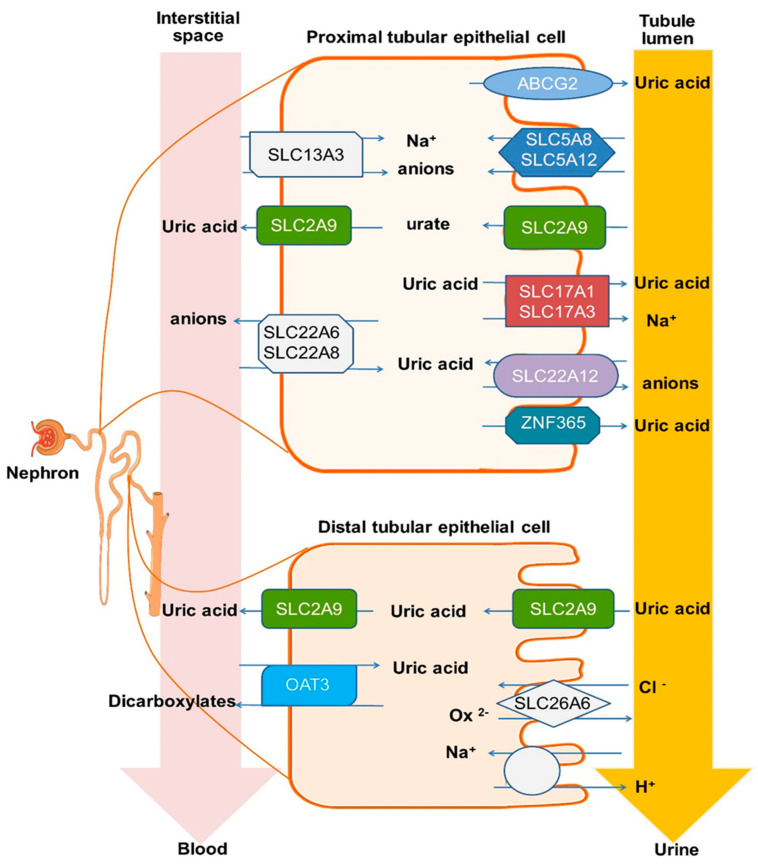
Diagrammatic presentation of the renal expression of six renal tubular uric acid (UA) transporters and their functions in the proximal and distal nephron. SLC2A9 (solute carrier family 2, glucose transporter member 9) is expressed in both the apical and basolateral membrane of proximal and distal tubular cells, and functions as a urate uniporter/urate reabsorption. SLC17A3 (solute carrier family 17 (organic anion transporter), member 3) is expressed on the apical membrane of the proximal tubular epithelium and transports urate and anionic compounds into the tubular fluid. SLC22A12 (solute carrier family 22 (organic anion/cation transporter member 12) or urate transporter 1 (URAT1) is expressed on the apical membrane of the proximal tubular epithelium and functions as a urate monocarboxylate exchanger to reabsorb urate from tubular lumen into the cytosol. SLC5A8 (solute carrier family 5, member 8) is expressed on the apical membrane of proximal tubular epithelium and functions as an electrogenic Na^+^ and Cl^−^ dependent sodium-coupled solute transporter/urate reabsorption. ABCG2 (ATP-binding cassette transporter G2) is expressed in both the apical and basolateral membranes of the proximal tubular epithelium and functions as a urate extrusion pump to aid in UA excretion. ZNF365 (zinc finger protein 365) is expressed on proximal tubular epithelial cells, has been associated with UA stones, and is proposed to contribute to UA excretion. The pre-selected renal UA-transporters play an important role in the pathophysiology of UA secretion and reabsorption, and urinary pH regulation in the kidney.

**Figure 2 ijms-23-10010-f002:**
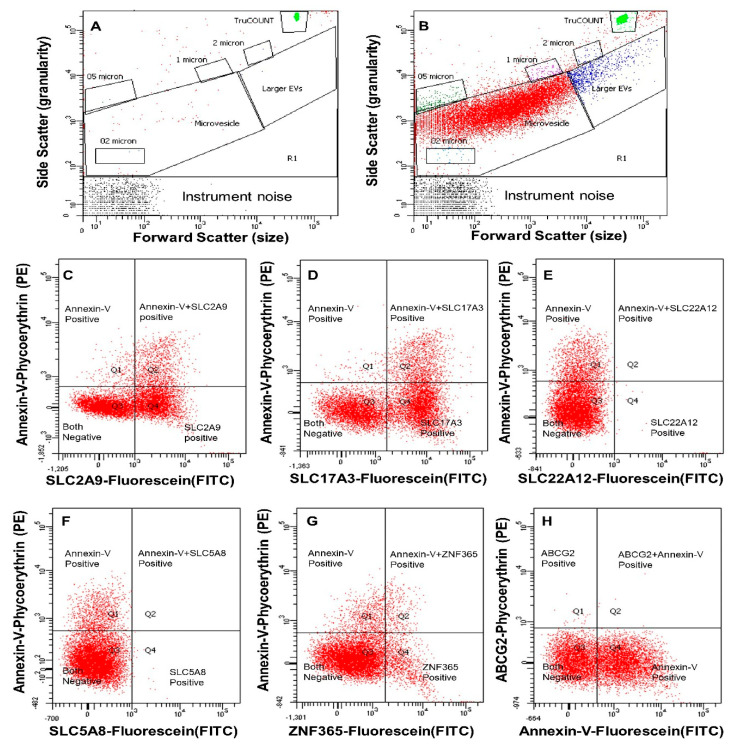
Representative scatter plots and fluorescence dot plots from the FACSCanto™ flow cytometer. (**A**) Example scatter plot of buffer with fluorophore-conjugated antibodies and calibration beads (no samples). (**B**) Example scatter plot of diluted urine plus appropriate fluorophore-conjugated antibodies and calibration beads. (**C**–**H**) Fluorescent dot plot (quadrants derived from microvesicle gate (contain major portions of extracellular vesicles) of (**B**)) showing fluorophore spectra for Annexin-V and SLC2A9 antibodies (**C**), SLC17A3 antibodies (**D**), URAT1 antibodies (**E**), ZNF365 antibodies (**F**), ABCG2 antibodies (**G**), and SLC5A8 antibodies (**H**).

**Figure 3 ijms-23-10010-f003:**
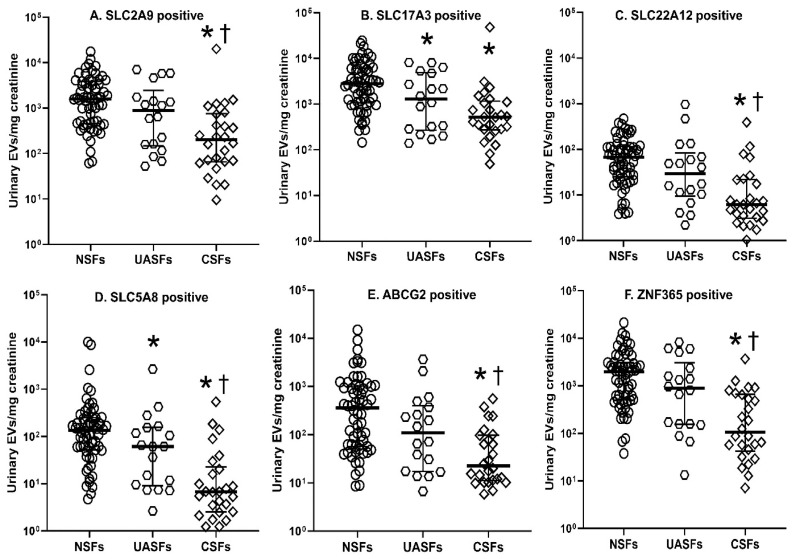
Urinary extracellular vesicles (EVs; microvesicles and exosomes) bearing six uric acid (UA)-transporters from UA stone formers (UASFs) and calcium stone formers (CSFs) and age-/sex-matched non-stone formers (NSFs). Urinary EVs were presented as the log_10_ of EVs/mg creatinine. * *p* < 0.05 UASFs or CSFs vs. NSFs; ^†^ *p* < 0.05 UASFs vs. CSFs (Wilcoxon rank sum test). The concentrations of SLC2A9, SLC17A3, and ZNF365-bearing EVs were greater than SLC22A12, SLC5A8, and ABCG2-expressing EVs in human urine.

**Table 1 ijms-23-10010-t001:** Baseline clinical characteristics of study participants.

Clinical Parameters	NSFs (n = 65)	UASFs (n = 18)	CSFs (n = 26)
Age (years)	63 (56, 73)	64 (55, 74)	65 (59, 74)
Female, n [%]	26 [40]	7 [39]	10 [38]
Body mass index (kg/m^2^)	27 (24, 30)	32 (27, 42) *	29 (26, 34)
Systolic blood pressure (mm Hg)	126 (111, 138)	130 (112, 143)	127 (113, 136)
Diastolic blood pressure (mm Hg)	73 (66, 82)	69 (58, 72) *	70 (61, 78)
Malabsorption syndrome, n [%]	3 [5]	2 [11]	2 [8]
Hypertension, n [%]	18 [28]	9 [50]	12 [46]
Diabetic mellitus, n [%]	11 [17]	9 [50]	8 [31]
Estimated GFR (mL/min/1.73 m^2^)	78 (62, 91)	67 (46, 76) *	62 (51, 76) *
**Blood biochemistry**			
Serum uric acid (mg/dL)	5.0 (4.0, 6.2)	6.7 (5.6, 7.0) *	5.8 (4.8, 6.5) *
Serum phosphorous (mg/dL)	3.3 (3.0, 3.7)	3.3 (2.9, 3.6)	3.6 (3.2, 3.8) *
Serum calcium (mg/dL)	9.3 (8.9, 9.6)	9.3 (9.2, 9.8)	9.5 (9.3, 9.7)
Serum creatinine (mg/dL)	0.9 (0.69, 1.1)	1.1 (0.9, 1.4) *	1.0 (0.9, 1.2) *
**Urine biochemistry**			
Urine pH	6.0 (6.0, 7.0)	5.7 (5.4, 6.1) *	6.0 (5.5, 6.2) *
Osmolality (mOsm)	528 (388, 763)	495 (346, 726)	409 (241, 580) *
Urine volume (mL/24 h)	1955 (1402, 2634)	2167 (1518, 2544)	2064 (1443, 2680)
Protein (mg/24 h)	26 (10, 42)	71.5 (22, 303) *	33 (16, 61)
Uric acid (mg/24 h)	376 (295, 581)	580 (431, 710) *	584 (456, 775) *
Phosphorus (mg/24 h)	516 (353, 826)	872 (714,1406) *	942 (713,1304) *
Calcium (mg/24 h)	139 (99, 200)	140 (105, 236)	255 (168, 327) *^,†^
Creatinine (mg/24 h)	939 (644, 1325)	1384 (1117, 1715) *	1573 (1235, 1977) *
CaOx (supersaturation, dG)	1.2 (0.8, 1.9)	1.5 (0.9, 1.9)	1.8 (1.2, 2.3) *
Uric acid (supersaturation, dG)	−0.7 (−6, 0.2)	0.8 (0, 3.9) *	0.8 (−1, 1.9) *

Continuous variables data were presented as median (interquartile range) and categorical variables data were reported as the number [percentage] (n, [%]) of non-stone formers (NSFs), uric acid stone formers (UASFs), and calcium stone formers (CSFs). Significant differences between groups were determined by Wilcoxon rank sum test. * *p* < 0.05 compared with NSFs; ^†^
*p* < 0.05 compared with UASFs. Abbreviations: dG, ∆Gibbs; GFR, glomerular filtration rate; CSFs, calcium stone formers; CaOx, calcium oxalate; NSFs, non-stone formers, UASFs, uric acid stone formers.

**Table 2 ijms-23-10010-t002:** Twenty-four-hour excretion of urinary extracellular vesicles (EVs; microvesicles and exosomes) bearing six uric acid (UA)-transporters from UA stone formers (UASFs) and calcium stone formers (CSFs) and age-/sex-matched non-stone formers (NSFs).

Uric Acid Transporter-Positive EVs Excreted (10^6^ per 24 h)	NSFs (n = 65)	UASFs (n = 18)	CSFs (n = 26)
SLC2A9 positive	1477 (477, 2872)	963 (342, 3089)	399 (167, 1450) *
SLC17A3 positive	2786 (1047, 4549)	2148 (507, 4977)	1305 (587, 2343) *
SLC22A12 positive	52 (24, 102)	41 (14, 119)	13 (6, 22) *^,†^
SLC5A8 positive	98 (48, 188)	129 (14, 257)	13 (7, 28) *^,†^
ABCG2 positive	183 (66, 763)	228 (31, 612)	47 (25, 187) *
ZNF365 positive	1404 (519, 2968)	1448 (336, 3401)	316 (71, 1146) *^,†^

Data were presented as median (25th, and 75th percentiles), EVs × 10^6^/24 h. * *p* < 0.05 UASFs or CSFs vs. NSFs; ^†^ *p* < 0.05 UASFs vs. CSFs (Wilcoxon rank sum test). The excretions of SLC2A9, SLC17A3, and ZNF365-bearing EVs were greater than SLC22A12, SLC5A8, and ABCG2-expressing EVs in human urine.

## Data Availability

The datasets used and/or analyzed during the current study were presented as individual data points from each study group in Figure 2 of this manuscript and are available from the corresponding author on reasonable request.

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
