# Peer review of "Differences of Uric Acid Transporters Carrying Extracellular Vesicles in the Urine from Uric Acid and Calcium Stone Formers and Non-Stone Formers"

_ijms, 2022, doi:10.3390/ijms231710010_

Round 1

Reviewer 1 Report

The authors study a group of uric acid stone disease cases (UASD) and controls (non-stone formers) by quantification of extracellular vesicles (EVs) by digital flow cytometer. The target proteins were identified via literature review as monogenic cause of UASD. The study is quite interesting and well presented. A few representative images of the digital flow cytometry data could improve the presentation and sell the manuscript better (see for example PMC4345020).

Minor:
In discussion: "serum uric acid concentration. . "

Reviewer 2 Report

Dear authors,

Many thanks with your responses, these clarified the research done.

Response:   As noted in the paper, we analyzed biobanked samples that were collected under two different IRB approved studies.  The calcium and uric acid stone formers were enrolled in a study of patients undergoing surgery for stone removal. The NSFs were from a population-based study comparing first time stone formers to non-stone former controls.  In both protocols subjects consented for use of their biobanked samples in future studies. In the current study, we sought to compare 6 specific urinary EV populations in patients with surgically-confirmed UA or  calcium stones and non-stone former controls. It is not unusual to combine and compare biobanked specimens in ancillary studies. Indeed, that is one envisioned future use for samples when they are biobanked in prospective studies. Attached is one example from our team [1] .   

Review comments:

It is necessary to introduce these observations in the Material and methods of the paper so that the future reader clearly understands that the study has been carried out with biobanked urine. In the text, this aspect is not clear, since it is indicated that the urine and serum samples were kept in a biobank and not obtained in a biobank. Please review all inconsistencies of Material and methods.

According to the journal's instructions for authors “Research Involving Human Subjects” and Helsinki declaration is necessary, according to point 23 of Helsinki declaration, an approval from the local institutional review board (IRB) or other appropriate ethics committee must be obtained before undertaking the research to confirm that the study meets national and international guidelines, also when using biobanked samples. If ethical approval is not required, authors must either provide an exemption from the ethics committee or are encouraged to cite the local or national legislation that indicates ethics approval is not required for this type of study.

Response: As stated in the manuscript, serum creatinine was measured by an enzymatic assay (Roche Diagnostics, Indianapolis, IN) (by definition not Jaffe, which is not an enzymatic assay).  We are not aware that 2 assays are described for serum creatinine. Urine creatinine was also measured by the enzymatic Roche assay, and we have added this detail. The typo “Isotype” has been corrected to “isotope”.  

Review comments:

Please include the determination of creatinine (underlined text) in urine determination in the Material and methods text:

“Blood and 24 hr urine samples (collected using toluene as a preservative) were obtained at the time of a Mayo Stone Clinic visit (CSFs and UASFs) or a study visit (NSFs). Serum (uric acid, calcium, phosphorus, creatinine) and urine (sodium, potassium, chloride, uric acid, calcium, mag nesium, phosphorus, creatinine, citrate (via citrate lyase), oxalate (via oxalate oxidase) biochemistries were analyzed using a Roche/Cobas C501 Autoanalyzer (Roche Diagnostics, Indianapolis, IN).”

Response: As requested, we have added the urinary EV concentration / µL urine data. Trends were quite similar when data for these 6 biomarker-expressing EVs were expressed as EVs/µl urine, with numbers being lowest in CSFs, while UASFs were intermediate and closer to NSFs (new Table 2). ”.  

Review comments:

The inclusion of Table 2 (the gold standard) modifies the results section and the entire paper discussion. There is also an incongruity between Figure 1 and Table 2. Table 2 being the gold standard, Figure 1 should be deleted since the information is lower than that provided by the gold standard.

The following is deduced from Table 2:

There is no significant difference in the EVs urinary elimination between uric acid stone formers and non-stone former for the EVs positive for the six transporters studied.

Three is significant decrease in the EVs urinary elimination between calcium stone formers and uric acid stone formers and between calcium stone formers and non-stone former for the EVs positive for the six transporters studied.

I have only reviewed the discussion section and I only present some inconsistencies detected as example:

Sentences:

“In this study, urinary EVs carrying SLC17A3 and SLC5A8 were significantly lower in UASFs compared NSFs.”

“Urinary EVs carrying SLC2A9, SLC22A12, SLC5A8, ABCG2, and ZNF365 were lower in CSFs than UASFs, while EVs  carrying SLC17A3 were not different between these two groups.”

These sentences contradict the gold standard expressed in Table 2: There is no significant difference in the EVs urinary elimination between uric acid stone formers and non-stone former for the EVs positive for the six transporters studied.

Sentences:

“Our current results suggest that renal tubular cells might respond to a specific type(s) of urinary crystals and tubular fluid compositions and release EVs containing specific UA transporters, and those changes may in turn reflect underlying UASD risk.”

“These EVs could also potentially serve as a signaling mechanism that mediates UA stone risk (positively or negatively).”

“For example, increasing tubular fluid uric acid concentration might down regulate proximal tubular cell secretion of EVs carrying SLC17A3 and SLC5A8 as a negative regulatory or unknown mechanism.”

How can authors support this argument? If There is no significant difference in the EVs urinary elimination between uric acid stone formers and non-stone former for the EVs positive for the six transporters studied.

The sentence:

The decrease in urinary excretion of EVs bearing UA transporters might reflect inhibition of EVs excretion from tubular cells by urinary uric acid crystals and low urinary pH, since the urine pH of CSFs and UASFs were both reduced compared to controls, with the low urinary pH more pronounced among the UASF group.

Which determination realized by the authors showing that urinary uric acid crystals were reduced compared to controls? From the results obtained, the authors cannot make this statement in the sentence.

The sentences:

EV surface expression of SLC2A9, SLC5A8, SLC17A3, SLC22A12, ABCG2, and ZNF365 might reflect altered UA transport capacity of tubular epithelial cells in a given individual, and could also signal other distally located tubular epithelial cells.

The populations of urinary EVs carrying SLC2A9, SLC17A3, and ZNF365 were greater than SLC22A12, SLC5A8 and ABCG2, and differed between UASFs and CSFs.

These sentences are not supported by table 2. There is no significant difference in the EVs urinary elimination between uric acid stone formers and non-stone former for the EVs positive for the six transporters studied.

The paper has been improved, but unfortunately still has many inconsistencies that can be avoided.

Round 2

Reviewer 2 Report

 Dear authors,

Many thanks with your responses. My comments in blue are incorporated under authors comments in red.

Response:   As noted in the paper, we analyzed biobanked samples that were collected under two different IRB approved studies.  The calcium and uric acid stone formers were enrolled in a study of patients undergoing surgery for stone removal. The NSFs were from a population-based study comparing first time stone formers to non-stone former controls.  In both protocols subjects consented for use of their biobanked samples in future studies. In the current study, we sought to compare 6 specific urinary EV populations in patients with surgically-confirmed UA or  calcium stones and non-stone former controls. It is not unusual to combine and compare biobanked specimens in ancillary studies. Indeed, that is one envisioned future use for samples when they are biobanked in prospective studies. Attached is one example from our team [1] .   

Review comments:

It is necessary to introduce these observations in the Material and methods of the paper so that the future reader clearly understands that the study has been carried out with biobanked urine. In the text, this aspect is not clear, since it is indicated that the urine and serum samples were kept in a biobank and not obtained in a biobank. Please review all inconsistencies of Material and methods.

According to the journal's instructions for authors “Research Involving Human Subjects” and Helsinki declaration is necessary, according to point 23 of Helsinki declaration, an approval from the local institutional review board (IRB) or other appropriate ethics committee must be obtained before undertaking the research to confirm that the study meets national and international guidelines, also when using biobanked samples. If ethical approval is not required, authors must either provide an exemption from the ethics committee or are encouraged to cite the local or national legislation that indicates ethics approval is not required for this type of study.

Authors comment

As is clearly stated in the manuscript these studies were approved by the IRB and the patients consented to this use of their samples.  To state otherwise would be untruthful, as these were consented studies and not exempt, which are different forms of approval.  Perhaps the review or is confused by the term biobank in our response.  These were not obtained from a separate biobank.  The investigators in this study were the ones that collected the samples and stored them in our laboratory for these studies as was approved by the IRB and patient has consented to.

Review comments

The information order in Material and Methods is not clear and this confuses me. It is necessary to introduce these observations also in the Material and methods of the paper. Please, if the Biobank is not the Mayo Clinic Biobank, avoid using this term because is confusing. The authors must inform the reader that they collected the samples and stored (indicate the conditions) them in their laboratory for make these studies as it was approved by the IRB and patient has consented to.

In this sentence “biobank center, Mayo Clinic” must be avoided if the Biobank is not the Mayo Clinic Biobank:

USD patients were recruited at the Mayo Clinic O’Brien Urology Research Center and cell free urine samples with toluene preservatives were stored at -80°C in biobank center, Mayo Clinic.

The order of information in Material and Methods is not clear. Please, rewrite the Material and Methods in a logic order according IJNS instructions for authors because is very confusing for the reader.

For example, as an idea for the authors, I rewrite the first paragraph of Material and Methods. The authors must forgive the reviewer, but the reviewer does not have the necessary information, I only incorporated the information extracted from the draft:

USD and NSF patients were recruited at the Mayo Clinic O’Brien Urology Research Center from participants in IRB: 08-006541; Epidemiology of nephrolithiasis and chronic kidney disease: Prospective Cohort; and IRB: 09-002083; CT and Urinary Correlates of Renal Stone Precursor Lesions. Blood and 24 h. urine samples (collected using toluene as a preservative) were obtained at the time of a Mayo Stone Clinic visit (CSFs and UASFs). All analyses were performed in the Clinical Laboratory Improvement Amendments (CLIA)-certified Mayo Clinic Renal Testing Laboratory, Rochester, MN. Aliquots of the 24h urine samples were centrifuged at 2100g/3000rpm for 10 min at 4°C to remove the cells and larger molecular weight protein aggregates before freezing and were stored at -80°C in our laboratory. After, the cell-free urine samples frozen (-80°C) were used for EV analysis. The participants gave written informed consent to use their urine samples.

Clinical data of the patients were obtained at the time of a Mayo Stone Clinic visit. Age, sex, body mass index, systolic blood pressure, diastolic blood pressure were obtained and the pathologic data about malabsorption syndrome, hypertension, diabetic mellitus and finally the glomerular filtration rate were estimated.

Authors comment

We strongly disagree that expressing urine EVs without normalizing to creatinine is the “gold standard”.  We have provided both normalized and non-normalized results and the trends and effect sizes for the associations were similar. Thus, we do not agree that the text and figures need to be extensively revised as Reviewer #2 suggests, since the results are fully presented in a very transparent way.

Review comments

I agree in part with the last authors comments. The review is not agreed with non-normalized results. The non-normalized results that is the gold standard is excretion of EVs in 24h. According to the first authors suggestion, the gold standard of excretion of EVs is EVs in 24 hours and this was accepted clearly by the reviewer. The reviewer expressed that the gold standard must be incorporated to the paper. But unfortunately, the authors incorporate a new Table about the number of EVs per urine volume avoiding incorporate the number of EVs /24 hours excretion, the gold standard suggested by the own authors. The production in 24 hours of EVs will be a clear indicator of the activity of the luminal membranes of the renal tubular cells in different pathological situations.

Finally, as general comment, the paper is interesting but there are some pitfalls that the authors must correct. The Material and methods must be rewritten in a logical way, in a correct order to be understand easily by the reader. Also is necessary include the gold standard of excretion of EV, suggested by the own authors. This information is necessary to know the production of EVs in the two pathologic situations respect to the physiologic situation.

.

*******************************************************************
